# Spinal Infections: An Update

**DOI:** 10.3390/microorganisms8040476

**Published:** 2020-03-27

**Authors:** Andreas G. Tsantes, Dimitrios V. Papadopoulos, Georgia Vrioni, Spyridon Sioutis, George Sapkas, Ahmed Benzakour, Thami Benzakour, Andrea Angelini, Pietro Ruggieri, Andreas F. Mavrogenis

**Affiliations:** 1First Department of Orthopaedics, National and Kapodistrian University of Athens, School of Medicine, 11527 Athens, Greece; andreas.tsantes@yahoo.com (A.G.T.); d.papado@yahoo.gr (D.V.P.); sp.sioutis@gmail.com (S.S.); gsapkas1@gmail.com (G.S.); 2Department of Microbiology, National and Kapodistrian University of Athens, School of Medicine, 11527 Athens, Greece; gvrioni@med.uoa.gr; 3Spinal Surgery Office, Clinique de l’ Archette, 45160 Olivet, France; benzak@hotmail.fr; 4Zerktouni Orthopaedic Clinic, 20000 Casablanca, Morocco; t.benzakour@gmail.com; 5Department of Orthopaedics and Orthopaedic Oncology, University of Padova, 35128 Padua, Italy; andrea.angelini83@yahoo.it (A.A.); pietro.ruggieri@unipd.it (P.R.)

**Keywords:** spine, abscess, spondylitis, spondylodiscitis, instrumentation

## Abstract

Spinal infection poses a demanding diagnostic and treatment problem for which a multidisciplinary approach with spine surgeons, radiologists, and infectious disease specialists is required. Infections are usually caused by bacterial microorganisms, although fungal infections can also occur. The most common route for spinal infection is through hematogenous spread of the microorganism from a distant infected area. Most patients with spinal infections diagnosed in early stages can be successfully managed conservatively with antibiotics, bed rest, and spinal braces. In cases of gross or pending instability, progressive neurological deficits, failure of conservative treatment, spinal abscess formation, severe symptoms indicating sepsis, and failure of previous conservative treatment, surgical treatment is required. In either case, close monitoring of the patients with spinal infection with serial neurological examinations and imaging studies is necessary.

## 1. Introduction

Spinal infections constitute a demanding diagnostic and treatment problem that in most cases necessitates a multidisciplinary approach with spine surgeons, radiologists, and infectious disease specialists. Infections are usually caused by bacterial microorganisms, although fungal infections can also occur. The nomenclature for spinal infections is complex and confusing since under the umbrella term “spinal infections” a heterogeneous group of infections is included (Table 1). This nomenclature correlates with several different aspects of the infection such as the causative pathogen, the underlying pathophysiology, or the involved part of the spinal column [1].

On the basis of the pathogen, bacterial spinal infections can be either pyogenic or granulomatous such as tuberculosis or brucellosis; any part of the spinal column can be affected correlating to different terms such as spondylodiscitis or discitis [1]. The infection can also spread inside the spinal canal, involving the dural sac or the epidural space, or to the paravertebral soft tissue. However, in most cases, the infection is not confined to only one anatomical compartment, it usually spreads and involves several different elements of the spinal column [2].

Spinal infections can also be classified based on the pathophysiology of the infection, and specifically based on the route of spread of the responsible pathogen [1]. Usually, infections originate from a distant site and via hematogenous spread, the microorganism reaches the spine. In other cases, the spinal column is infected through continuous spread from an adjacent infection. Last, infections can develop due to direct inoculation of the culprit microorganism during surgery or following local trauma (surgery or injury). The risk for postoperative infection following spinal surgery varies because it depends on many factors, importantly the type of the spinal procedure; the overall incidence of postoperative spinal infection is estimated from 0% to 18% [3]. In the last scenario the infection is also called secondary, as opposed to primary spinal infection that occurs without any previous history of trauma [3]. This article aims to summarize and critically examine the current evidence for spinal infections in a comprehensive review that the curious reader may find interesting and educative.

## 2. Εpidemiology

The overall incidence of spinal infections is approximately 2.2/100,000 per year [4,5]. The most common type of spinal infection is primary pyogenic spondylodiscitis, in which a bacterial microorganism infects the vertebral body and the intervertebral disc via hematogenous spread [6]. For reasons that are not fully elucidated, spondylodiscitis is twice as common in men [7]. Although spinal infections can affect patients of any age starting from infancy, vertebral spondylodiscitis most commonly occurs in adults and has a predilection for patients aged >50 years; intravenous drug abusers are a younger age group with high susceptibility to spinal infections [8,9]. There are several factors that can possibly explain the higher prevalence of spinal infections in older people including immunosuppressive medications, intravascular devices, and other forms of implants that are more common in older patients [10].

The most frequently involved spinal segment is the lumbar spine (58%), followed by the thoracic spine (30%), and the cervical spine (11%) [6]. Hematogenous sacral infection is rare [11,12,13]. In most cases, sacral osteomyelitis results from pressure ulcers, trauma, surgery, or through contiguous spread from a pelvic infection [11,12]. The spinal infection can extend posteriorly and result in epidural or subdural abscess, or even meningitis, while lateral spread can result in psoas, retroperitoneal, subphrenic, paravertebral, retropharyngeal, and mediastinal abscesses. The most common culprit for epidural or psoas muscle abscesses is Gram-positive bacteria [13]. Other parts of the vertebrae that can be infected include the facet joints and the spinous processes [14].

As with any infection, any disease that compromises the immune system such as diabetes mellitus, HIV/AIDS infection, malignancy, renal failure, hepatic cirrhosis, and malnutrition poses a significant risk factor for a patient to experience a spinal infection [15,16,17,18,19,20,21]. Medications such as immunosuppressive agents following organ transplantation or long-term corticosteroids also increase the risk for spinal infections. Other significant risk factors for spinal infections include previous spinal surgery and presence of intravascular or orthopaedic implants. Last, intravenous drug abusers are susceptible to infections not only because of their deteriorated immunologic defense, but also because of the high risk for injection site infections from infected needles or vasculitis [4].

## 3. Pathogenesis

Bacteria can reach the spine and infect the spinal column via the following three routes: (1) hematogenous spread from a remote site, (2) direct external inoculation after trauma (injury or surgery), and (3) dissemination from a contiguous tissue [1]. Hematogenous spread is the most common route for vertebral osteomyelitis in children and adults [22,23,24,25,26,27,28,29]. Generally, any condition that results in circulation of microorganisms into the blood stream (bacteremia) such as surgery or more benign events such as tooth brushing or venipuncture, can lead to hematogenous spondylodiscitis. Infection in the urinary tract, often following genitourinary procedures, is the most common source of transient bacteremia and subsequent spinal infection [22,23,24,25,26,27,28,29]. Other common potential primary sources for hematogenous spondylodiscitis include gastrointestinal infections, otitis media, oral cavity infections, infective endocarditis, skin and soft tissue infections, respiratory tract infections, and infected intravenous catheter sites. In almost 50% of cases, the primary source of infection is not identifiable [30]. Secondary spinal infections due to direct inoculation can occur after spinal surgery or minimally invasive spinal procedures such as chemonucleolysis or discography, or after penetrating trauma in the spinal area [31,32,33,34,35,36,37,38]. Contiguous spread to the spine from an infection in an adjacent structure such as the aorta, the esophagus, or the bowel have been also reported [39,40].

Considering the vasculature of the spine, the same vascular pedicle bifurcates supply two adjacent vertebral end plates; therefore, in most cases infection involves two adjacent vertebral bodies and their intermediate disc [1]. The vertebral end plates are the first to be infected, and at a subsequent time the infection propagates to the adjacent disc or to the vertebral body. Furthermore, the slow blood flow in these vessels, the lack of valves, and convolution of the arterial or venous supply make vertebral column more susceptible to develop an infection in patients with bacteremia [1]. In children, as opposed to adults, the infection can spread more easily since the blood vessels in the end plates extend to supply the intervertebral discs as well [41].

## 4. Clinical Presentation

The most common complaints of patients with spinal infection is back or neck pain, depending on the location of the infection. Occasionally, pain radiates to the lower limbs, genital area, groin, or even to the abdomen. Pain usually aggravates at night and can be severe enough to awaken the patient. Although it is very difficult to differentiate the common mechanical back pain from a spinal infection, spondylodiscitis should always be investigated in febrile patients with back pain [1]. However, although fever is a common symptom, it is not always present; approximately 50% of the patients with pyogenic spine infections, and even in a higher percentage of patients with fungal, mycobacterial, and brucellar infections are afebrile [3].

On physical examination, inspection should assess for any scar indicating a previous surgery or trauma. In addition, any deformity such as kyphosis or scoliosis should be noted. On palpation, paravertebral muscle spasm can be felt, while marked tenderness with percussion over spinous processes of the infected spinal segment is a more consistent finding, reported to be present in 75% to 95% of the cases. However, rarely, in patients with Charcot arthropathy and paraplegics is pain minimal or absent [1]. Approximately 30% of the patients present with neurological symptoms including paresthesia and muscle weakness [5]. In the case of epidural abscesses resulting from posterior extension of the infection to the epidural space, neurological deficits including radiculopathy, weakness or paralysis, and paresthesia is more common (Figure 1) [42]. Due to the low specificity of symptoms and signs, diagnosis can be delayed for several weeks or even months; therefore, increased awareness and clinical suspicion is necessary.

## 5. Diagnosis

The diagnostic approach (Figure 2) for the patients with spinal infections should begin with a complete medical history and physical examination during which possible risk factors for infection must be always investigated and identified. Initial blood work-up should include white blood cell count (WBC) and inflammatory markers such as erythrocyte sedimentation rate (ESR) and C-reactive protein (CRP). CRP is considered to have the highest sensitivity as compared with other blood tests (98%), while elevated ESR is observed in 75% of cases [1,21,27]. However, a sole elevation of CRP can be misleading, thus, an elevated CRP in conjunction with elevated ESR or WBC and clinical symptoms consistent with spinal infection (such as back pain and fever) are more indicative of a spinal infection. CRP is also a useful marker of response to antimicrobial therapy because it normalizes rapidly after successful treatment, as opposed to ESR that can remain elevated for a long time after clinical improvement [1]. WBC is a less useful laboratory parameter as compared with CRP and ESR, because it is normal in up to 55% of patients with spinal infections [21]. *Brucella* serology is mandatory in cases with signs and symptoms indicating brucellosis, exposure to a potential source, or in endemic areas [27]. Interferon-gamma (IFN-*γ*) release assay should be also performed in patients with risk factors for tuberculosis such as immigrants, low socioeconomical status, and endemic countries. This assay can be also useful for exclusion of active spinal tuberculosis, due to its high (95%) negative predictive value [43].

Imaging evaluation of the patients with spinal infections should include radiographs of the spine and magnetic resonance (MR) imaging with contrast medium administration [44]. Typical MR imaging findings of spondylodiscitis include (1) hypointense vertebral bodies and disc with loss of endplate definition in T1-weighted images, (2) hyperintense vertebral bodies and disc with loss of endplate definition in T2-weighted images or STIR images, and (3) contrast enhancement of the vertebral body and disc. Imaging should include the entire spine in order to assess the extension of the infection and to exclude any adjacent or skip lesions. Bone scintigraphy with technetium or labelled leucocytes is not routinely indicated because of their low sensitivity and specificity; gallium scans can have a role because if the result is negative, osteomyelitis is unlikely [45]. Newer tracers for bone scintigraphy such as indium-111 labeled (111In) biotin and streptavidin have been recently introduced [46]. Streptavidin accumulates in sites of inflammation and infection, while 111In-biotin has a high affinity for streptavidin; a sensitivity, specificity and diagnostic accuracy of 94%, 95%, and 94%, respectively has been reported with the Streptavidin/111In-biotin scan for spinal infections [46]. Other new tracers include the technetium Tc-99m-ubiquicidin-derived peptide that has a high affinity to sites with viable bacterial growth, in addition to radiolabeled antifungal tracers for differentiation of fungal from bacterial infections [47]. At present, the Current Infectious Diseases Society of America (IDSA) guidelines recommend the use of 18F-FDG-PET/CT only in cases that MR imaging is contraindicated [27].

Although with a relatively low sensitivity, at least two sets of blood cultures for aerobic and anaerobic bacteria should be obtained [2]; in approximately 60% of spinal infections, the implicated pathogen can be identified in blood cultures [1]. Cultures on specific media for *Mycobacterium tuberculosis* and fungi should be requested as well, especially when there is a suspicion for tuberculosis exposure or indicating imaging signs [7]. A CT-guided percutaneous needle aspiration biopsy is paramount to confirm the diagnosis and isolate the responsible microorganism, provided that prompt surgical treatment is not required such as in cases with progressive neurological symptoms due to spinal cord compression or spinal canal abscesses [48]. However, the sensitivity and specificity of CT-guided needle biopsy is lower than believed; in a recent metanalysis, a sensitivity of 52.2% (95% CI, 45.8–58.5) for CT-guided percutaneous needle aspiration biopsy for the diagnosis of spinal infections was reported [48]. Recent studies reported a higher pathogen detection rate of fine-needle aspiration with combined superimposed MR and CT imaging [49,50]. Aspirate of the disc or paraspinal soft tissue lesions should be obtained, and samples should be sent for microbiology, histology, and ideally for nucleic acid amplification testing (NAAT). Kim et al. showed that the pathogen detection rate of soft tissue sampling is 2.28 times higher as compared with bone tissue [51]. NAAT such as polymerase chain reaction (PCR) testing is useful in cases of negative aerobic and anaerobic cultures in patients who have already taken antibiotics. NAAT testing is also useful in cases of unusual microorganisms or slow growing bacteria such as *Mycobacterium tuberculosis* or *Coxiella burnetii* [52,53]. Histopathology is useful, especially when cultures are negative. The presence of leucocytes in histological sections indicates bacterial infection, while granuloma formations indicates specific pathogens such as *Mycobacterium Tuberculosis* or *Brucella*. When clinical symptoms and imaging are typical for vertebral osteomyelitis, and blood cultures are positive for a common pathogen such as *Staphylococcus aureus*, a needle biopsy is not required [11].

Although it is a common practice to withhold antibiotics prior to biopsy, recent studies have shown that antibiotics prior to biopsy do not result in lower culture sensitivity [54,55]. Additionally, in cases of critically ill patients such as in septicemia, empirical antibiotic therapy should start promptly without any delay [27]. In cases of negative blood cultures with two consecutive negative needle biopsies, open biopsy is recommended. Open biopsy for tissue sampling yields a pathogen detection rate of 68% to 93% [56].

## 6. Differential Diagnosis

In some cases, symptoms of spinal infections can be very similar to those of other spinal pathologies, and thus differential diagnosis is warranted. Especially in elderly patients with no fever, misdiagnosis is very common. Spinal infections should be differentiated from spinal tumors, spinal stenosis, herniated nucleus pulposus, and simple muscle strains. MR imaging is very useful in differentiating spinal stenosis and herniated nucleus pulposus from infection since the imaging findings are unique for each pathology. Apart from imaging studies, clinical evaluation can be also helpful to differentiate mechanical pain (with or without neurological symptoms) from infections. There are some distinctive features between the typical mechanical back pain and the pain due to a spinal infection; in patients with mechanical back pain due to muscle sprain, pain aggravates with upright posture and with daily activities; in contrast, in patients with spinal infections the pain is constant regardless of the activity and can aggravate at rest and night [1]. Differential diagnosis between spinal tumors and infections is more difficult and complicated. Frequently, the clinical manifestation, MR imaging findings, and laboratory tests are inconclusive. In these cases, biopsy is required as it is the only reliable method to distinguish these two conditions.

## 7. Microbiology

Although there is a wide range of bacteria that can cause spinal infections, in most cases these infections are caused by a single microorganism rather than from multiple pathogens [45]. In those few poly-microbial infections (<10% of cases), the spine is usually affected through contiguous spread [30]. In hematogenous spinal infections, in almost 50% of the cases, the source of infection can be identified [30]; this is most commonly the genitourinary tract (17%) followed by endocarditis (12%), skin and soft tissue (11%), gastrointestinal (5%), and respiratory system (2%) [57].

In general, three major groups of microorganisms cause spinal infections, i.e., bacteria (pyogenic infections), fungi, and very rarely parasites. In the past, *Mycobacterium tuberculosis* was considered to be the most common cause for spinal infections, with some studies reporting tuberculosis in 50% of the patients with spinal infections [28]. In recent times, however, the microbiology of spinal infections has changed; currently, most spinal infections are pyogenic, with *Mycobacterium tuberculosis* being the isolate in <25% of the cases in certain areas [57]. Among all pathogens, *Staphylococcus aureus* is the most common microorganism, responsible for 20% to 84% of all spinal infections, while approximately 5% to 20% of spinal infections are caused by Streptococci and Enterococci, and in <4% of the cases by anaerobic microorganisms [58,59,60]. *Enterobacteriae spp*. are considered to be the culprit in 7% to 33% of pyogenic infections, with *Escherichia coli* being the most common microorganism of this group, followed by *Proteus* and *Klebsiella*. The latter microorganisms are common causes of urinary tract or gastrointestinal infections, especially in diabetic or immunosuppressed patients [16,61]. Salmonellosis, caused by a pathogen of the same species, is frequently seen in children with sickle-cell disease [62]. Vertebral osteomyelitis caused by *Pseudomonas aeruginosa* is common in intravenous drug users, although *Staphylococcus aureus* remains the most common microorganism in this group of patients as well [63,64]. In patients with implants, such as patients with prosthetic joint replacements, the risk of *Staphylococcus epidermidis* spinal infection is higher [61]. Other coagulase-negative *Staphylococci* such as *Staphylococcus viridans* can cause low-grade infections due to their lower virulence. Fungal infections are very rare, and most commonly affect immunocompromised patients [1]. Another microorganism that has been implicated in orthopedic infections is *Cutibacterium Acnes*, formerly known as *Propionibacterium Acnes*. *Cutibacterium acnes* is a Gram-positive anaerobic-aerotolerant bacillus and is part of the normal human flora as it resides in skin follicles, in the eye mucosa, and in the oral cavity or in the rest of the gastrointestinal tract. The percentage of bone infections caused by *Cutibacterium acnes* varies, ranging from 2% to 18% [2]. Its association with orthopaedic infections is not entirely clear. The reasons for that are that in the past it was considered to be a culture contaminant due its normal presence in the skin, while the infection from this pathogen is in many cases subclinical and delayed. On the one hand, there are increasing data, especially in shoulder surgeries, that point towards its causative role in postoperative infections. On the other hand, there are only few data regarding *Cutibacterium acnes* in spinal infections. Interestingly, there are studies reporting positive cultures for *Cutibacterium acnes* in disc material of herniated discs, implicating this microorganism in the development of disc herniation [2]. There are slight differences regarding the microbiology of spinal infections in children; in this age group, *Staphylococcus aureus* and *Streptococcus spp.* are the most common pathogens, while another common isolate for spondylodiscitis or discitis in children is *Kingella kingae* [65]. The reason for the more frequent report of *Kingella kingae* as an etiologic factor for osteoarticular infections in children is because now this pathogen can be more easily detected. Although this Gram-negative organism is difficult to be isolated in several body fluids, the use of aerobic blood culture vials or NAAT testing such as PCR, lead to identification of this bacterium as a cause of many infections in ages 6 months to 4 years, a group in whom traditional culture is very frequently negative.

Pathogens for spinal infections also vary depending on geographic location. The incidence of infection from *Brucella* is considerably higher in Mediterranean and Middle Eastern countries, with the spine involved in 6% to 12% of patients with brucellosis [66]. Similarly, although spinal infections from *Echinococcus* are extremely rare, they can be seen occasionally in endemic regions such as countries with a warm climate such as South America, central Asia, China, Australia, and Africa [67].

## 8. Conservative Treatment Antibiotics

The goals of conservative treatment for patients with spinal infections is eradication of the infection and pain relief, while spinal stability is preserved, and neurological dysfunction is prevented. In approximately 90% of the cases, conservative treatment is successful in achieving these goals [68]. Conservative treatment includes appropriate antibiotic therapy and pain medications combined with spinal brace immobilization and physical therapy. Bed rest is usually recommended, especially during the initial period in the context of pain relief and prevention of spinal deformity. Usually, this period should last approximately one to two weeks, or until pain improvement. Following this period, ambulation with a spinal brace is recommended. In rare cases, where the infection has significantly spread and affected a major part of the anterior spinal column, a longer period of bed rest, up to six weeks, could be necessary [1]. However, a long immobilization period is associated with increased morbidity, especially in elderly patients, in whom the risk for certain complications such as pressure ulcers, pulmonary embolism, or respiratory tract infections is high [3]. In cases of cervical spine infections, immobilization can be obtained with a neck collar or with a halo-vest in more advanced cases with extensive bone destruction. In the thoracic spine, an extension brace that can prevent a kyphotic deformity of the spine is recommended. Accordingly, a lower thoracolumbar or lumbosacral brace is indicated when the thoracolumbar or the lumbar spine are involved [1].

Despite the high rate of success with conservative treatment, in patients with imaging evidence of progressive destruction of the spinal column or with progressive neurological deficits, conservative treatment should be discontinued, and surgical treatment should be performed. Similarly, when clinical improvement is not achieved after an initial period of conservative treatment for approximately six weeks, surgical treatment should be considered as well [69,70]. It is desirable to begin antibiotics therapy after isolation of the responsible pathogen [55], except if a patient’s condition is deteriorating [71].

There is a wide range of antibiotics for spinal infections. In critically ill patients or when cultures are negative, a dual-agent empirical therapy including a third-generation cephalosporin or fluoroquinolones, plus clindamycin or vancomycin is recommended. Until isolation of the causative pathogen, empirical treatment should include clindamycin/vancomycin/flucloxacillin + cefepime/ciprofloxacin/ceftriaxone to cover a wide spectrum of potential pathogens (Table 2). Subsequently, antibiotics therapy should be modified based on the results of cultures. An anti-staphylococcal penicillin or a first-generation cephalosporin is the recommended antibiotic regimen when a methicillin-sensitive *Staphylococcus* is isolated. Otherwise, when a methicillin-resistant microorganism is identified, which is common with *Staphylococcus aureus* pathogens, a glucopeptide antibiotic such as vancomycin or teicoplanin is indicated. Alternative agents include quinupristin-dalfopristin or linezolid. When *Streptococcus spp.* are identified, penicillin G is the recommended antibiotic agent [72]. In cases of Gram-negative microorganisms, a second- or third-generation cephalosporin, or a quinolone can be administered, whereas for anaerobic pathogens metronidazole or clindamycin are recommended.

The recommended regimen for spinal tuberculosis includes multiple agents given that the risk of antibiotic resistance is significantly high [73]. The recommended four-agents regimen includes isoniazid, rifampicin, ethambutol, and pyrazinamide. For spinal infections caused by *Brucella*, a dual agent antibiotic treatment with doxycycline and streptomycin (or gentamicin) is recommended. In rare cases of fungal infections, a long-term treatment with azole or amphotericin B should be administered [72]. Despite the general recommendations regarding the antibiotic agents, it should be noted that in each case the antibiotic regimen must be individualized based on the cultures and susceptibilities of the isolated pathogens to certain antibiotic agents.

The optimal duration of antimicrobial therapy for spinal infections is highly debatable [45]. There is a wide variety of recommendations [1,44], with most authors recommending a period of four to six weeks [44], while others recommending a longer period of up to three months [68,74]. In a recent randomized controlled trial enrolling 359 patients with pyogenic spondylitis, the authors concluded that a six-week antibiotic treatment was similarly successful and safe to a 12-week treatment [75]. In that study, 176 patients were allocated to the six-week antibiotic group, while 175 patients were allocated to the 12-week group. In both groups, approximately 90% of patients were successfully treated and clinically cured, while the rate of adverse events was similar; death incidence was 8% for the six-week group vs. 7% for the 12-week group, drug intolerance was 7% vs. 5%, and neurological complications occurred in 4% vs. 2% of the patients [75].

Antibiotics therapy can be discontinued when clinical improvement with resolution of symptoms is evident, and the inflammatory markers are normalized [3]. However, in spinal infections caused by *Brucella* or *Mycobacterium tuberculosis,* a longer antibiotic treatment is required. Although the exact optimal duration is unclear, patients with *Brucella* infections should be administered antibiotics for three to six months [76], and patients with spinal tuberculosis should be administered even longer; usually 9 to 12 months of antibiotic treatment is recommended for eradication of the mycobacterial infection and prevention of recurrence using a four-agents regimen for the first two months, limited to two agents for the rest of this period [71,77,78]. There are no specific guidelines regarding the duration of treatment for fungal infections, and a more individualized approach is recommended in these cases taking into consideration the side effects of antifungal antibiotics and the clinical response to treatment [79]. The duration of treatment is also recommended to be longer in cases of undrained abscesses or in patients with infected spinal implants (Figure 3) [74]. Similar to adults, there are no specific guidelines about the duration of therapy in children. A common protocol for children includes intravenous antibiotics for one to three weeks until clinical and laboratory improvement is evident, followed by oral antibiotic for another one to three weeks [1].

Intravenous administration of antibiotics is recommended for the first two to four weeks [69], although there are contradicting data regarding the efficacy of switching to oral antibiotics in less than four weeks [80,81]. Oral antibiotics with high bioavailability such as fluoroquinolones allow a safer switch from intravenous treatment, while other agents with lower bioavailability such as beta-lactam antibiotics are not recommended to be used as sole oral antibiotics following intravenous treatment [74]. Recent studies have suggested that an early switch to oral antibiotics has similar results to long term intravenous antibiotics [82,83]. A large multicenter randomized study (OVIVA study) compared six-week intravenous antibiotics to six-week oral antibiotics for the treatment of osteomyelitis, including patients with spondylodiscitis. The primary end point in this study was definite treatment failure at one year. The authors found that there was non-inferiority with oral treatment [82]. Another retrospective study including patients with vertebral osteomyelitis treated surgically as compared with less than or equal to three weeks intravenous postoperative antibiotic course (followed by four weeks oral therapy) to a longer (greater than three weeks) intravenous antibiotic postoperative treatment and reported similar results in low risk patients; however, in patients with paraspinal abscess formation or positive blood cultures, short term therapy resulted in higher rate of recurrence [83].

It has been proposed that a 50% weekly decrease in CRP levels indicates satisfactory clinical improvement [16]. In contrast, if fever and pain do not resolve after a four-week course of antibiotics, and CRP levels remain persistently high (>30 mg/l) failure of the conservative treatment is the most likely scenario and surgical treatment should be considered [74].

## 9. Surgical Management

Indications for surgery include failure of conservative treatment with no resolution of symptoms, septic status, spinal instability, spinal canal abscesses (Figure 4), paravertebral abscess >2.5 cm, and spinal cord or nerve root compression with progressive neurological deficits (Figure 5, Table 3) [45]. Spinal deformity such as kyphosis or scoliosis can be the end result of a spinal infection that in most cases need also to be addressed surgically [84]. Although in most cases surgical treatment is not needed, almost 50% of patients with spinal infection will undergo some sort of surgery [58].

Usually, adequate surgical debridement and decompression is feasible only through an anterior approach, since in most cases the vertebral bodies and disks are affected. However, a posterior approach is required in cases of spinal canal abscess. In many cases, surgical debridement results in spinal instability, therefore, an additional stabilization procedure with bone grafting (either autograft or allograft) or spinal instrumentation is required. Moreover, anterior fusion only is not adequate in some cases, and a combined anterior-posterior approach with supplementary posterior stabilization can be required, especially in multi-segmental involvement [84]. Although large strut autografts are ideal for fusion due to their high rate of consolidation, they are associated with donor site morbidity. Therefore, instrumentation such as with titanium cages, as well as transpendicular screws and rods are commonly used. However, due to the risk of bacterial colonization of the implants, the issue of instrumentation in an infected environment is still debatable [84,85,86]. *Staphylococcus* can colonize hardware creating thick biofilm. It has been shown that there is limited biofilm formation on titanium due to a lack of porosity, therefore, it is a preferred material in such cases, and titanium cages have shown favorable results [60]. Prerequisites for spinal instrumentation at the time of debridement include thorough debridement and concomitant antibiotic therapy. Authors of a recent study reported 0% rate of recurrence in patients with spinal infections treated with fusion and instrumentation at the time of the debridement [85]. However, this was a small case series of nine patients, while patients received a long-term postoperative antibiotic course, up to 12 months. Another study reported a higher re-operation rate after decompression only vs. decompression and stabilization with spinal instrumentation [86].

Patients with postoperative spinal infections and infected implants often require irrigation and debridement with implant removal [58]. Treatment management differs based on the chronicity of the postoperative infection. In patients with early infections (less than three months) with present spinal instrumentation, removal of the instrumentation is not recommended in order to avoid spinal destabilization in an infected bed [87,88]. Although loose bone grafts should be debrided during surgery, stable grafts that are adherent to native bone should be left in place.

Late postoperative infections are usually recommended to be treated with implant removal. First, given that solid fusion has been achieved, complete debridement is not feasible because areas such as spinal anchorage points or the region directly under the rods are relatively inaccessible without removing the instrumentation. Moreover, late postoperative infections are caused by microorganisms that usually form biofilms, such as coagulase-negative *Staphylococci* or *Cutibacterium acnes*. Di Silvestre et al. found that retention of the instrumentation during debridement of delayed spinal infections can result in 50% probability of persistent infection [89]. However, it is not always easy to assess whether osseous fusion has occurred, therefore, the benefit of eradicating the biofilm should be weighed against the risk of destabilizing the spine by removing fixation. Moreover, in chronic cases with long fusions, there is the risk of fracturing the fusion mass or lose alignment during implant removal. If osseous fusion has not achieved, bone grafts (autograft or allograft) can be used for bony fusion, without an increase in postoperative infection rate [89]. A treatment algorithm for spinal infection is presented in Figure 6.

## 10. Conclusions

The diagnostic approach of patients with spinal infections should include blood workup, blood or CT-guided needle cultures and histology, and imaging evaluation with radiographs and MR imaging. In addition to confirmation of the infection, the diagnostic algorithm should always aim to identify the source of the infection, and to establish a microbiological diagnosis. Provided that patients are neurologically intact and with no severe signs of infections, many authors propose withholding antibiotics, if empirically administered previously, in order to optimize culture sensitivity. However, when patients present with sepsis, empiric antibiotics should start immediately and not be withheld for cultures.

Most patients with spinal infections diagnosed in early stages can be successfully managed conservatively with antibiotics, bed rest, and spinal braces; a commonly used empirical antibiotics regimen includes vancomycin and a third-generation cephalosporin (such as cefepime) or a fluoroquinolone to cover MRSA and Gram-negative organisms, but other antibiotic regimens with similar coverage can be used as well. In cases of gross or pending instability, progressive neurological deficits, failure of conservative treatment, spinal abscess formation, severe symptoms indicating sepsis, and failure of previous conservative treatment surgical treatment is required.

Close monitoring of the patients with spinal infection with serial neurological examinations and imaging studies is necessary. Although the main goals of management and the overall treatment protocols have not changed over time, antibiotic therapy and techniques for spinal stabilization have significantly evolved and improved. Titanium implants currently are the optimal hardware for stabilization following debridement, due their favorable properties resulting in less biofilm formation.

## Figures and Tables

**Figure 1 microorganisms-08-00476-f001:**
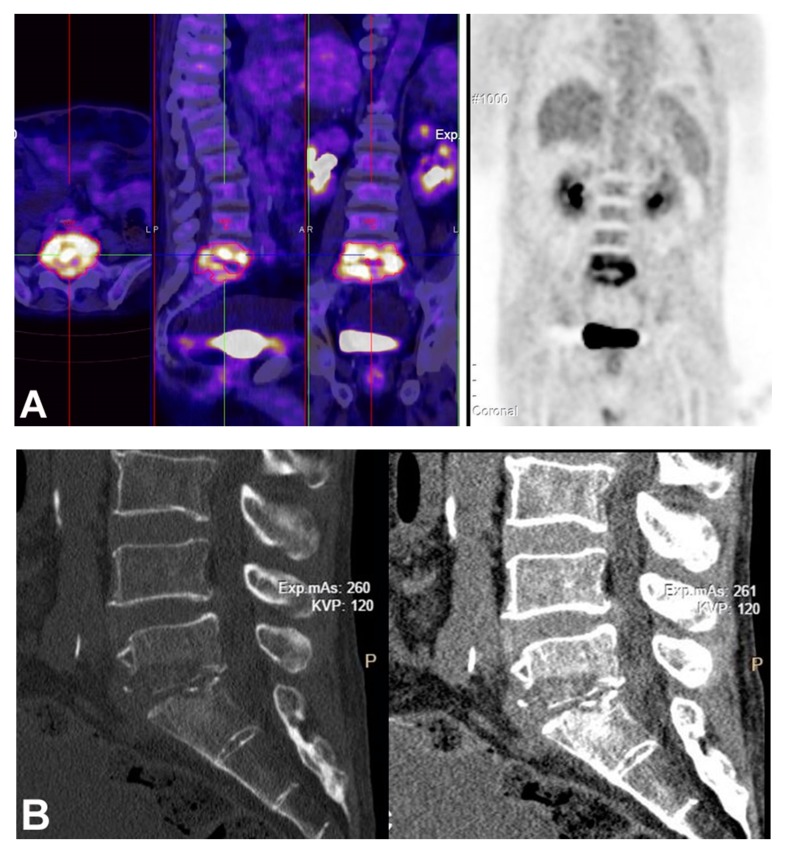
A 64-year-old man with HBV-related hepatic cirrhosis, and L5-S1 MRSA spondylodiscitis. (**A**) PET/CT shows diffuse uptake at the L5-S1 level (SUV, 4.06); (**B**) Sagittal CT scans show complete destruction of the L5-S1 intervertebral disc and erosion of L5 and S1 vertebra; (**C**) T1-weighted magnetic resonance (MR) imaging shows abscess formation at the L5-S1 level. He was treated with antibiotics and surgical decompression.

**Figure 2 microorganisms-08-00476-f002:**
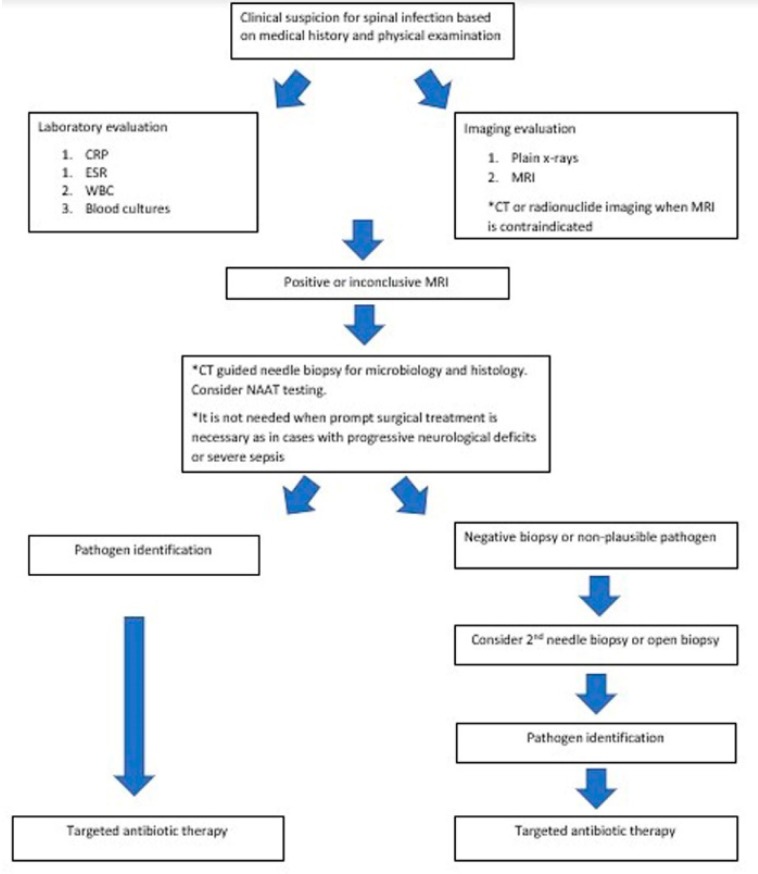
Diagnostic algorithm for spinal infections.

**Figure 3 microorganisms-08-00476-f003:**
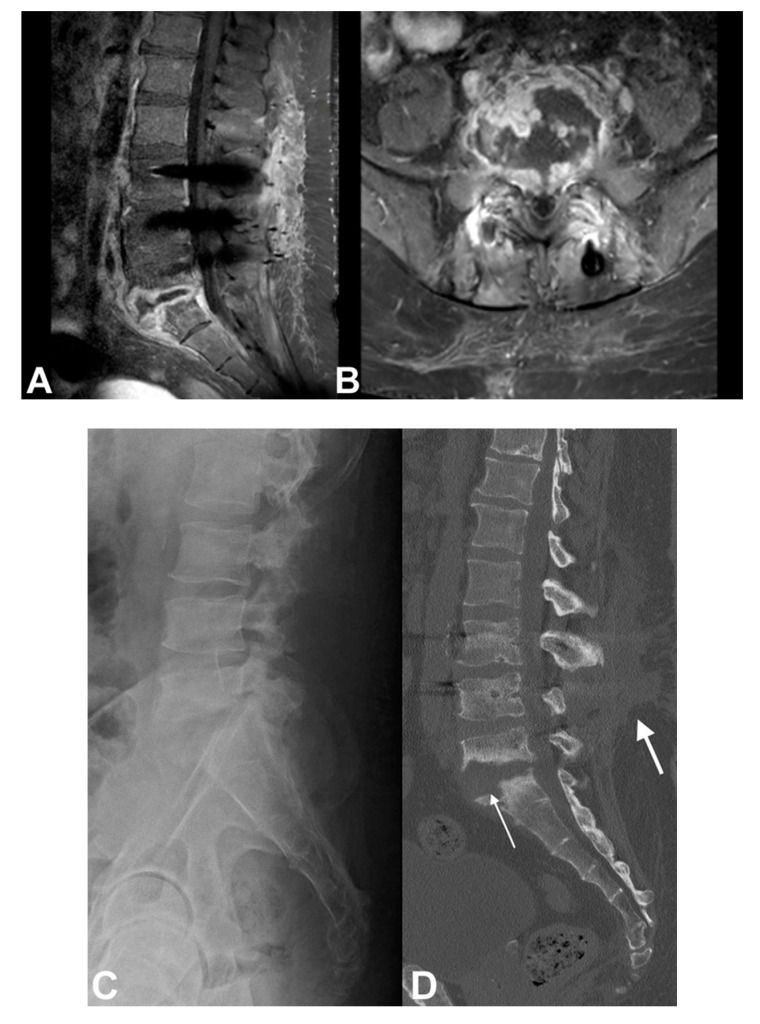
A 42-year-old woman with insulin dependent diabetes mellitus and obesity, and MRSA infection 1 year after L3-L5 laminectomy and spinal instrumentation. (**A**) Sagittal and (**B**) axial MR imaging show abscess formation and implants loosening at the site of instrumentation. She was treated with surgical debridement and implants removal followed by a 6-month antibiotics regimen; (**C**) Lateral radiograph and (**D**) sagittal CT scan of the lumbar spine show erosion of L5-S1 vertebrae (thin arrow) and extensive scar tissue formation (thick arrow).

**Figure 4 microorganisms-08-00476-f004:**
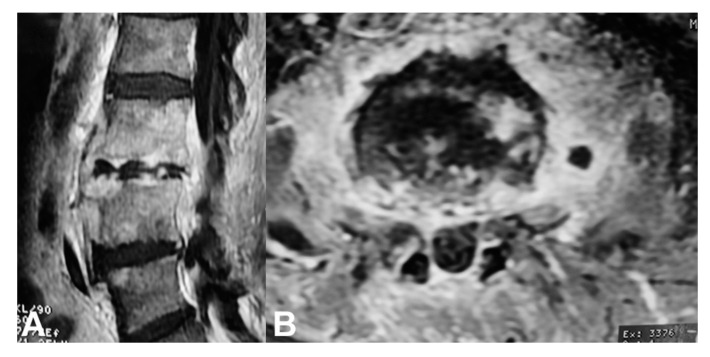
A 62-year-old man with L3-L4 MRSA spondylitis a couple of weeks after an infected olecranon bursitis. (**A**) Sagittal and (**B**) axial MR imaging show extensive destruction of the L3 and L4 vertebrae and abscess formation extending to the spinal canal. He was treated with surgical decompression and abscess drainage followed by a 6-month antibiotics regimen.

**Figure 5 microorganisms-08-00476-f005:**
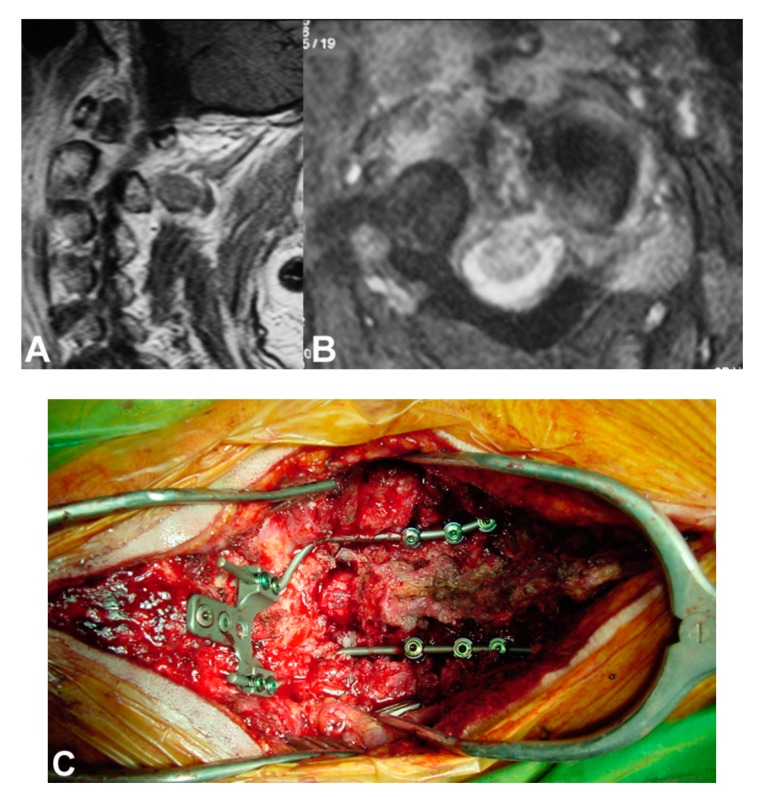
A 70-year-old man with a C1-C2 MRSA spondylitis and epidural abscess formation. (**A**) Sagittal and (**B**) axial MR imaging show erosion of the C1-C2, destruction of the odontoid process and abscess epidural formation. Because of progressive neurological deficits (tetraplegia) he was treated with (**C**) antero-posterior decrompression and craniocervical fusion followed by a 6-month antibiotics regimen.

**Figure 6 microorganisms-08-00476-f006:**
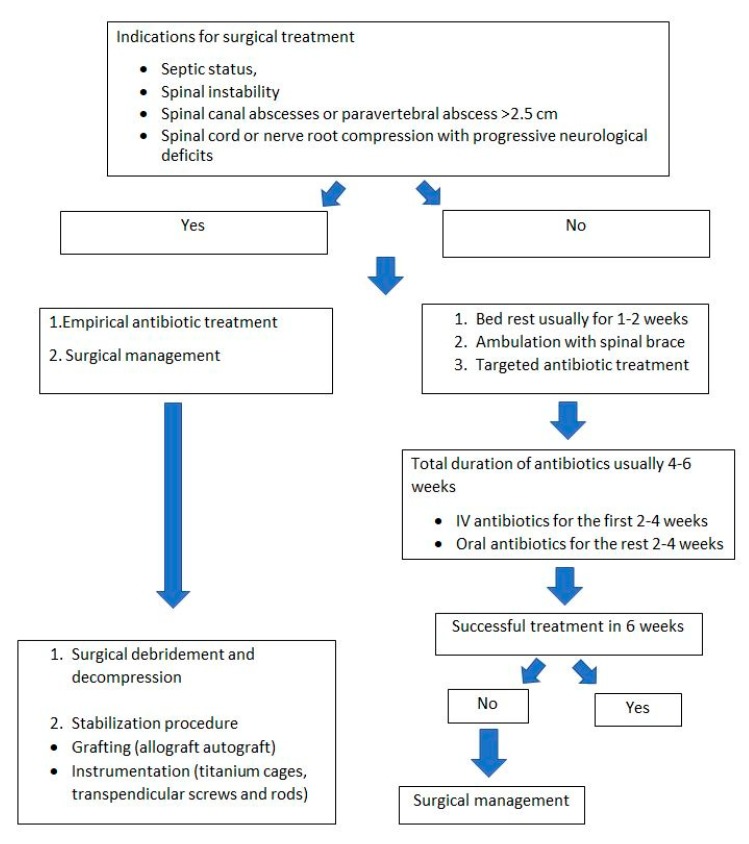
Treatment algorithm for spinal infections.

**Table 1 microorganisms-08-00476-t001:** Terminology of the spinal infections.

Term	Site of Infection	Features
Discitis	Intervertebral disc	Common in children
Spondylitis	Vertebral end plate and vertebral body	Similar to osteomyelitis, usually seen at early stage of infection in adults
Spondylodiscitis	Disc and adjacent vertebral body	Most common form of spinal infection
Septic facet joint	Facet joints	Hematogenous spread to the facet joints, increasingly diagnosed over the past years
Epidural abscess	Epidural space	Rarely seen as isolated abscess, contiguous spread of infection into the medullary canal

**Table 2 microorganisms-08-00476-t002:** Antibiotics for initial and empirical treatment.

Agents	Bacterial Susceptibility
Clindamycin, Flucloxacillin, Vancomycin, Teicoplanin	Staphylococcus, Streptococcus, MRSA
Ciprofloxacin, Cefepime	Gram-negative bacteria
Chloramphenicol, Amoxicillin+Clavulanic acid, Meropenem/Imepenem,	Anaerobic bacteria

**Table 3 microorganisms-08-00476-t003:** Indications for surgical treatment.

Indications
Failure of conservative treatment after 6–8 weeks
Sepsis
Progressive neurological dysfunction
Spinal instability
Epidural abscess

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
