# Peer review of "Spinal Infections: An Update"

_microorganisms, 2020, doi:10.3390/microorganisms8040476_

Round 1

Reviewer 1 Report

The authors described in a very nice review update of spinal infections. After a complete introduction, they described epidemiology, pathogenesis, clinical presentation, diagnosis, microbiology and conservative treatment of spinal infections. This review reflects the complexity of spinal infections, with a high variety of microorganisms and the difficulties of treatment without irreversible damages for patients.

Anyway, a few sentences do not sound clear to me in the abstract (lines 16-17 and lines 19-20, repeat lines 44-45).

Do authors know the percentage of risk to develop infections after surgery (risk of direct inoculation)?

Sentence lines 117-118 need to be developed.

Line 139, CRP has to be associated with other symptoms and markers.

In the "Microbiology" part, authors should talk about Cutibacterium acnes (and not anymore Propionibacterium acnes as mentioned line 379). Why are there differences in identified microorganisms between ages? It could be mentioned.

In table 2, the authors could talk about antibiotics against anaerobic strains.

Authors should pay attention to the quality of figures. "gram" should be written "Gram" with an upper case. (line 399).

Author Response

Comments and Suggestions for Authors

Reviewer 1

 The authors described in a very nice review update of spinal infections. After a complete introduction, they described epidemiology, pathogenesis, clinical presentation, diagnosis, microbiology and conservative treatment of spinal infections. This review reflects the complexity of spinal infections, with a high variety of microorganisms and the difficulties of treatment without irreversible damages for patients.

Thank you for your comments Reviewer 1, we agree. We hope that we have revised our manuscript appropriately, as per yours and the Reviewer’s 2 comments below. Changes in the text are highlighted in red; the figures are highlighted in yellow.

  1. Anyway, a few sentences do not sound clear to me in the abstract (lines 16-17 and lines 19-20, repeat lines 44-45). Do authors know the percentage of risk to develop infections after surgery (risk of direct inoculation)?

Thank you for your comments Reviewer 1, we agree. These sentences have been clarified in the abstract, and the percentage of the risk to develop infections after surgery has been added, as requested.

Abstract: Spinal infection poses a demanding diagnostic and treatment problem for which a multidisciplinary approach with spine surgeons, radiologists and infectious disease specialists is required. Infections are usually caused by bacterial microorganisms, although fungal infections can also occur. The most common route for spinal infection is through hematogenous spread of the microorganism from a distant infected area.

Introduction, ultimate paragraph: Last, infections can develop due to direct inoculation of the culprit microorganism during surgery or following local trauma (surgery or injury). The risk for postoperative infection following spinal surgery varies because it depends on many factors, importantly the type of the spinal procedure; the overall incidence of postoperative spinal infection is estimated between from 0% to 18% [3]. In the last scenario the infection is also called secondary, as opposed to primary spinal infection that occurs without any previous history of trauma [3].

  1. Sentence lines 117-118 need to be developed.

Thank you for your comment Reviewer 1, we agree. These sentences were expanded, as recommended.

Clinical Presentation, paragraph 2: On physical examination, inspection should assess for any scar indicating a previous surgery or trauma. Also, any deformity such as kyphosis or scoliosis should be noted. On palpation, paravertebral muscle spasm can be felt, while marked tenderness with percussion over spinous processes of the infected spinal segment is a more consistent finding, reported to be present in 75% to 95% of the cases.

  1. Line 139, CRP has to be associated with other symptoms and markers.

Thank you for this comment as well Reviewer 1, we also agree. Your comment has been addressed at the revised manuscript, as recommended.

Diagnosis, paragraph 1: CRP is considered to have the highest sensitivity compared to other blood tests (98%), while elevated ESR is observed in 75% of cases [1,27,43]. However, a sole elevation of CRP can be misleading, thus an elevated CRP in conjunction with elevated ESR or WBC and clinical symptoms consistent with spinal infection (such as back pain and fever) is more indicative of a spinal infection. CRP is also a useful marker of response to antimicrobial therapy because it normalizes rapidly after successful treatment, as opposed to ESR that can remain elevated for long after clinical improvement [1].

  1. In the "Microbiology" part, authors should talk about Cutibacterium acnes (and not anymore Propionibacterium acnes as mentioned line 379). Why are there differences in identified microorganisms between ages? It could be mentioned.

Thank you for your comments Reviewer 1, we agree. Yours comments have been addressed at the revised manuscript, as recommended. A short discussion about Cutibacterium acnes has been added in the Microbiology part, the right term of this microorganism has been corrected in the line 379, and the issue of different microorganisms between ages has been further elaborated. Thank you for your comments Reviewer 1.

Microbiology, paragraph 2: Another microorganism that has been implicated in orthopedic infections is Cutibacterium Acnes, formerly known as Propionibacterium Acnes. Cutibacterium acnes is a Gram positive anaerobic-aerotolerant bacillus and is part of the normal human flora as it resides in skin follicles, in the eye mucosa and in the oral cavity or in the rest of the gastrointestinal tract. The percentage of bone infections caused by Cutibacterium acnes varies, ranging from 2% to 18% [2]. Its association with orthopaedic infections is not entirely clear. The reasons for that are that in the past it was considered to be a culture contaminant due its normal presence in the skin, while the infection from this pathogen is in many cases subclinical and delayed. However, there are increasing data especially in shoulder surgeries that point towards its causative role in postoperative infections. On the other hand, there are only few data regarding Cutibacterium acnes in spinal infections. Interestingly though, there are studies reporting positive cultures for Cutibacterium acnes in disc material of herniated discs, implicating this microorganism with development of disc herniation [2]. There are slight differences regarding the microbiology of spinal infections in children; in this age group, Staphylococcus aureus and Streptococcus spp. are the most common pathogens, while another common isolate for spondylodiscitis or discitis in children is Kingella kingae [66]. The reason for the more frequent report of Kingella kingae as an etiologic factor for osteoarticular infections in children is because now this pathogen can be more easily detected. Although this Gram-negative organism is difficult to be isolated in several body fluids, the use of aerobic blood culture vials or NAAT testing like PCR, lead to identification of this bacterium as a cause of many infections in ages 6 months to 4 years, a group in whom traditional culture is very frequently negative.

Discussion, ultimate paragraph:…… Moreover, late postoperative infections are caused by microorganisms that usually form biofilms, such as coagulase-negative Staphylococci or Cutibacterium acnes……

  1. In table 2, the authors could talk about antibiotics against anaerobic strains.

Thank you Reviewer 1, we agree. Your comment has been addressed in Table 2, as recommended.

  1. Authors should pay attention to the quality of figures. "gram" should be written "Gram" with an upper case. (line 399).

Thank you for your comments Reviewer 1, we agree. Editing was done to the revised manuscript, as necessary.

 Reviewer 2 Report

This review article entitled “Spinal infections: an update” by Andreas G. Tsantes is a comprehensive review about spinal infections. The descriptions on this paper are correctly based on the previous papers. This paper is well-organized and will be of interest to readers of microorganisms. I have only a few concerns mentioned below.

Comments

1. The article is well-organized and written in details, however, the present form lacks the information about differential diagnosis of spinal infection. In clinical setting, we should differentiate spinal infection from mimics including spinal tumors, degenerative diseases, etc. The authors are encouraged to add such description to the current form.

2.    Flow chart of diagnosis and treatment would be useful for readers to apply this review article to the treatment of patients. The authors are encouraged to add flow charts to the current form.

Author Response

 Reviewer 2

 This review article entitled “Spinal infections: an update” by Andreas G. Tsantes is a comprehensive review about spinal infections. The descriptions on this paper are correctly based on the previous papers. This paper is well-organized and will be of interest to readers of microorganisms. I have only a few concerns mentioned below.

Thank you for your comments Reviewer 2. We hope that we have revised our manuscript appropriately, as requested by yours and Reviewer’s 1 comments above.

  1. The article is well-organized and written in details, however, the present form lacks the information about differential diagnosis of spinal infection. In clinical setting, we should differentiate spinal infection from mimics including spinal tumors, degenerative diseases, etc. The authors are encouraged to add such description to the current form.

Thank you for your comment Reviewer 2, we agree. A new section has been added in the revised manuscript regarding the differential diagnosis of spinal infections.

Differential Diagnosis: In some cases, symptoms of spinal infections can be very similar to those of other spinal pathologies thus differential diagnosis is warranted. Especially in elderly patients with no fever, misdiagnosis is very common. Spinal infections should be differentiated from spinal tumors, spinal stenosis, herniated nucleus pulposus and simple muscle strains. MR imaging is very useful in differentiating spinal stenosis and herniated nucleus pulposus from infection since the imaging findings are unique for each pathology. Apart from imaging studies, clinical evaluation can be also helpful to differentiate mechanical pain (with or without neurological symptoms) from infections. There are some distinctive features between the typical mechanical back pain and the pain due to a spinal infection; in patients with mechanical back pain due to muscle sprain, pain aggravates with upright posture and with daily activities; in contrast, in patients with spinal infections the pain is constant regardless of the activity and may aggravate at rest and night [1]. Differential diagnosis between spinal tumors and infections is more difficult and complicated. Many times the clinical manifestation, MR imaging findings and laboratory tests are inconclusive. In these cases, biopsy is required as it is the only reliable method to distinguish these 2 conditions.

  1. Flow chart of diagnosis and treatment would be useful for readers to apply this review article to the treatment of patients. The authors are encouraged to add flow charts to the current form.

Thank you for your comments Reviewer 2, we agree. In the revised manuscript we added 2 figures (new figures 2 and 6) with the flow charts for the diagnosis and treatment algorithm for spinal infections. The figures were renumbered. Thank you for your comments Reviewer 2.

Figure 2. Diagnostic algorithm for spinal infections.

Figure 6. Treatment algorithm for spinal infections.